# Advanced Intrusion Detection Combining Signature-Based and Behavior-Based Detection Methods [†]

**Hee-Yong Kwon [1]**, **Taesic Kim [2]**, and **Mun-Kyu Lee [1,*]**

[1] Department of Electrical and Computer Engineering, Inha University, Incheon 22212, Korea; heeyong.kr@gmail.com

[2] Department of Electrical Engineering and Computer Science, Texas A&M University-Kingsville, Kingsville, TX 78363, USA; taesic.kim@tamuk.edu

[*] Correspondence: mklee@inha.ac.kr; Tel.: +82-32-860-7456

[†] This paper is an extended version of our work presented at ICNGC 2021, entitled "Hee-Yong Kwon; Taesic Kim; Mun-Kyu Lee. A Hybrid Intrusion Detection Method for Industrial Control Systems", in which we presented a hybrid intrusion detection method. In this full version, we conducted additional experiments to fine-tune various parameters and anomaly detection criteria. Consequently, we further improved the performance of anomaly detection. In addition, we also demonstrated the efficiency of the proposed method in terms of execution time.

**Abstract:** Recently, devices in real-time systems, such as residential facilities, vehicles, factories, and social infrastructure, have been increasingly connected to communication networks. Although these devices provide administrative convenience and enable the development of more sophisticated control systems, critical cybersecurity concerns and challenges remain. In this paper, we propose a hybrid anomaly detection method that combines statistical filtering and a composite autoencoder to effectively detect anomalous behaviors possibly caused by malicious activity in order to mitigate the risk of cyberattacks. We used the SWaT dataset, which was collected from a real water treatment system, to conduct a case study of cyberattacks on industrial control systems to validate the performance of the proposed approach. We then evaluated the performance of the proposed hybrid detection method on a dataset with two time window settings for the composite autoencoder. According to the experimental results, the proposed method improved the precision, recall, and F1-score by up to 0.008, 0.067, and 0.039, respectively, compared to an autoencoder-only approach. Moreover, we evaluated the computational cost of the proposed method in terms of execution time. The execution time of the proposed method was reduced by up to 8.03% compared to that of an autoencoder-only approach. Through the experimental results, we show that the proposed method detected more anomalies than an autoencoder-only detection approach and it also operated significantly faster.

**Keywords:** autoencoder; neural network; cybersecurity; industrial control system; intrusion detection

## 1. Introduction

Recently, a wide variety of devices have been connected to communication networks in real-time control systems, such as residential facilities, vehicles, factories, and social infrastructure. These devices or systems were physically controlled in the past, but administrators can now manage and control them efficiently and remotely because of this increased network connectivity. However, this convenience can also result in critical concerns and challenges regarding cybersecurity. In this study, we considered the cybersecurity of industrial control systems (ICSs). ICSs are used to control industrial processes, such as manufacturing, product handling, production, and distribution. The majority of these systems monitor complex industrial processes and critical infrastructure that deliver power, water, transportation, manufacturing, and other essential services. Owing to this fundamental importance, ICSs are considered as major targets for cybercriminals. For example, many attacks have been conducted targeting supervisory control and data acquisition (SCADA)

systems. In 2010, Iran's nuclear power plant was attacked with the Stuxnet worm [1] and Ukraine's Chernobyl nuclear power plant was attacked with Petya ransomware in 2016 [2]. In 2020, SolarWinds, Honda Motors, and the University of Vermont Health Network were attacked with SUNBURST malware [3], Ekans ransomware [4], and Ryuk ransomware [5], respectively. Recently, in 2021, Acer was attacked with ransomware by an affiliate of REvil, also known as Sodinokibi [6], and Colonial Pipeline was attacked with ransomware by the Darkside cybercriminal group [7]. In particular, attackers who compromise the system security of cyber-physical systems (CPSs) can potentially manipulate actuators, inject forged or compromised data, and install malware to cause malfunctions that damage or destroy the system by penetrating an ICS via a communication network. Furthermore, the rapid implementation of the Internet of Things (IoT) in ICSs has significantly increased the attack surfaces of typical systems. Due to these security concerns and challenges, the identification of potential threats and the development of defense mechanisms to mitigate cyberattacks targeting ICSs are critical to leverage the proliferation of Industry 4.0 transformations. Therefore, cybersecurity is becoming an essential component of contemporary ICSs.

A wide variety of network security applications have been developed either to prevent attacks on ICSs before they are launched or to simply defeat standard attacks by closing the associated vulnerabilities. However, adversaries can perform attacks using vulnerabilities that have not been previously disclosed publicly, i.e., zero-day attacks. Therefore, to protect systems from attack, administrators cannot rely solely on network security solutions. To enhance ICS security, the development of robust and effective anomaly detection systems that are based on the live monitoring of operational system states is required. Intrusion detection systems (IDS), which were designed to detect attacks by identifying potentially harmful anomalies in sensor data or actuator behavior, are among the most efficient security measures available. Typically, IDSs are categorized into two types based on their detection methods: signature- and behavior-based methods. The signature-based approach detects anomalies by comparing system data to the features of known attacks, whereas the behavior-based approach detects and analyzes malicious or unusual patterns of behavior. In general, as the amount of anomalous data is significantly less than that of normal data, many anomaly detection methods extract features from normal data and use them to detect anomalous behavior. Recently, autoencoder-based models have attracted considerable attention as unsupervised methods that only require normal data to perform training [8–12]. However, the use of autoencoder-based methods involving complex neural networks can be an excessive approach for some obvious attacks that could have been detected by simple statistical measures. Therefore, a lightweight signature-based method can also be adopted to reduce the burden of the autoencoder by filtering obvious attacks using predefined statistical rules.

The contributions of this study are as follows:

- We propose a hybrid anomaly detection method that combines signature- and behavior-based methods to improve detection performance. For the signature-based detection, we used the standard deviations computed from normal data as the classification criteria;
- We evaluate the detection performance of the proposed method and present experimental results demonstrating that it outperformed the existing autoencoder-only method on the secure water treatment (SWaT) dataset. In addition, we compare the performance of the proposed method to those of previous detection methods that applied various machine learning approaches;
- We evaluate the execution time of the proposed method and demonstrate that it significantly accelerated the detection task compared to the previous autoencoder-only method. We proved the efficiency of the proposed method using a generalized numerical analysis.

The present work is an extended version of a paper presented at ICNGC 2021 [13]. The new contributions of this extended version are threefold. First, we considered two time window settings for the composite autoencoder to evaluate its performance, only one of

which was considered in [13]. Next, we analyzed the influence of various filtering bounds on the proposed method and further improved its detection performance by selecting the optimal threshold. Finally, we demonstrated the practical advantages of the proposed method through the measurements of its execution time, which showed that it performed faster than the existing method.

## 2. Related Works

### 2.1. Network Intrusion Detection

Depending on the installation location, IDSs can be classified as network-based IDSs (NIDSs) and host-based IDSs (HIDSs) [14]. An NIDS is installed at a specific point in the network, such as a gateway, where it can observe the network packets of a target system. It then detects an attack, intrusion or abnormal behavior targeting the system by analyzing the network packets generated by multiple devices in the system. In contrast, an HIDS is individually installed on each device constituting the target system. It then detects attacks by analyzing the status of the corresponding device. However, typical devices, such as sensors or actuators equipped in ICSs, are too severely resource-constrained to effectively perform as HIDSs. Therefore, we focus on NIDSs as a more realistic approach for ICS.

There has been extensive research on anomaly detection in the literature regarding NIDSs. Liu et al. proposed a detection method based on isolation forests [15]. They efficiently detected anomalies based on the natural observation that anomalies are easier to isolate from others than from normal data points. In [16], the authors proposed a clustering method based on Gaussian mixture models for denial of service (DoS) attack detection. Experiments were performed to identify the differences between a normal operation data point and the DoS-affected operation data points. In 2018, LinkedIn provided a lightweight anomaly detection library for time series data [17]. For given time series data, the library provides anomaly detection results and a time window during which an anomaly might have occurred. In addition, it helps to find correlation coefficients for two time series datasets. In 2021, Toldinas et al. proposed an NIDS using multistage deep learning image recognition [18]. They transformed network traffic features into four-channel (red, green, blue, and alpha) images and classified the images using a pre-trained ResNet50 model. Another study [19] proposed a detection method based on long short-term memory (LSTM). They used stacked LSTM networks for anomaly detection in time series data and validated the method using four real-world datasets. Another study [20] in 2016 proposed an encoder–decoder scheme for anomaly detection (EncDec-AD) that was based on LSTM. The method encodes a multi-sensor time series input to a vector representation and then decodes this vector representation to the original input data. Lee et al. proposed a zero-positive machine learning system called Greenhouse that does not use anomalous data for training [21]. In the study, the authors used differences between predicted and observed values and labeled the observed data as an anomaly when the difference exceeded a threshold.

For ICS security, Marti et al. used a one-class support vector machine (OCSVM) to detect anomalies in turbomachinery [22]. In that study, noisy, unreliable, and inconsistent data were pruned for efficient anomaly detection using a time series segmentation algorithm. When the difference between the input and predicted data was significant, alerts were sent. Filonov et al. proposed a fault detection method for multivariate industrial time series data [23]. The model architecture included two stacked LSTMs and it was validated using a mathematical model of part of a real gas–oil plant. In 2019, Kim et al. proposed an anomaly detection method using a sequence-to-sequence model (seq2seq) [24] and validated its performance using the SWaT dataset [25]. This method encodes the input data to a latent vector and predicts the future values of the input by decoding the latent vector. Another study [14] proposed a CNN-based anomaly detection method for payloads of ICS network traffic. They converted packet payloads into images and then performed detection processes on single packet and packet sequence bases.



## 2.2. Network Intrusion Datasets

To support active ICS intrusion detection research, various network intrusion datasets have been released to the public. KDD Cup 1999 is a dataset provided by MIT Lincoln Labs [26]. This dataset provides nine weeks of raw transmission control protocol (TCP) data dumps for a local area network (LAN) and includes a wide variety of intrusions that are simulated in a military network environment, including DoS, unauthorized access from a remote machine, unauthorized access to local superuser privileges, surveillance, and other probing. Lemay and Fernandez provided a dataset that includes malicious and non-malicious Modbus traffic for SCADA networks [27]. The dataset was generated in a SCADA sandbox to provide a dataset that is more similar to the real environment. In addition, they used electrical network simulators and employed real attack tools in the Modbus networks. Another study [28] provided datasets that were generated from two laboratory-scale systems: a gas pipeline and a water storage tank. The authors captured network flow records using a network data logger to monitor the Modbus traffic and process the measurement features. A set of 28 attacks was used to provide malicious data, which were grouped into reconnaissance, response injection, command injection, and DoS. Another study [29] shared a dataset of a laboratory-scale gas pipeline system. The dataset includes labeled network transactions that were generated from the testbed, encompassing normal situations and 35 cyberattacks. Shin et al. published a hardware-in-the-loop (HIL)-based augmented ICS security (HAI) dataset (version 1.0) in 2020 [30]. The authors generated HAI 1.0 from a testbed composed of a GE turbine, an Emerson boiler, and FESTO water treatment systems centered on an HIL simulator. The dataset was collected over 15.5 days of continuous operation, where the data of 10 days recorded only normal operations and the remaining 5.5 days included 38 attacks. In addition, data from 59 points in the testbed were gathered and the attacks were labeled using four attack tags. However, the SWaT dataset [25] is one of the most referenced datasets in the literature of ICS NIDSs [11,24,31–33]. The SWaT dataset was used in this study. The SWaT dataset is explained in detail in the following subsection.

## 2.3. Water Treatment ICS

In this study, we validated the performance of the proposed method using the secure water treatment (SWaT) dataset [25], which comprises data that are representative of those collected by a water treatment ICS. The SWaT dataset was generated and is provided by iTrust, of the Singapore University of Technology and Design, as a publicly available cyber-physical system (CPS) dataset. They constructed a testbed reflecting a real-world environment for water treatment systems, from which they collected the data.

Figure 1 presents the overall process of the SWaT testbed, which consists of six stages numbered from P1 to P6. The details of each stage are as follows:

- P1: **(Water storage)** Raw water is collected and stored in a tank;
- P2: **(Chemical dosing)** When the quality of the water is not within acceptable limits, chemical dosing is performed;
- P3: **(Fine filtration)** Undesirable materials are removed using fine filtration membranes;
- P4: **(Dechlorination)** The remaining chlorine is largely destroyed using ultraviolet lamps;
- P5: **(Reverse osmosis)** Inorganic impurities are reduced using a reverse osmosis system;
- P6: **(Ready for distribution)** Potable water is stored in a specialized tank and is then ready for distribution.

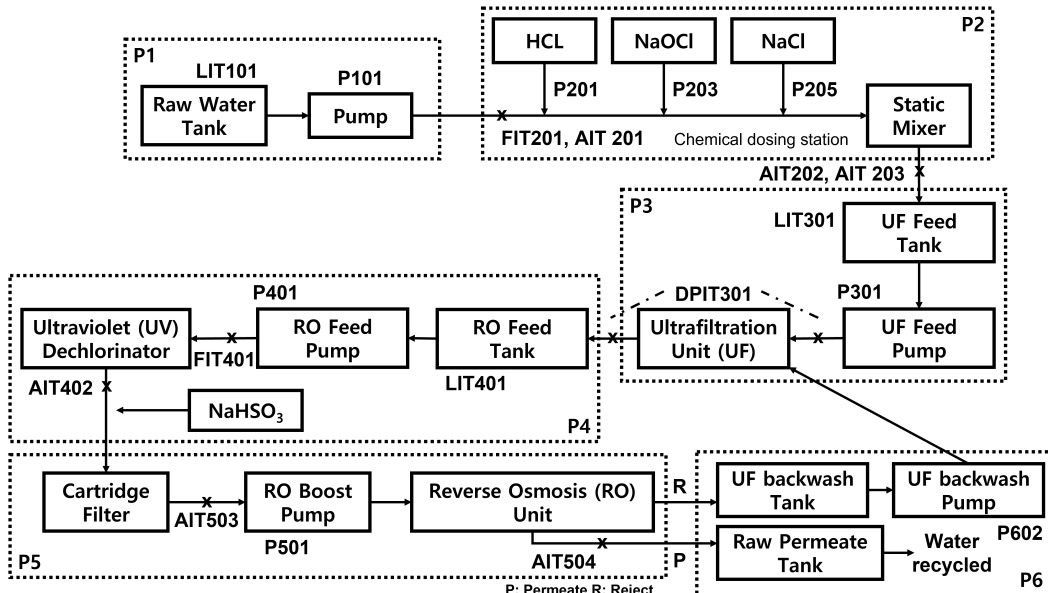

**Figure 1.** The whole process of the SWaT testbed [25]. Reproduced with permission from Springer Nature Customer Service Centre GmbH: Springer, Cham, Critical Information Infrastructures Security 2016, "A dataset to Support Research in the Design of Secure Water Treatment Systems," Jonathan Goh, Sridhar Adepu, Khurum Nazir Junejo, and Aditya Mathur, © Springer International Publishing AG 2017 (2017).

Throughout the whole process, data from 24 sensors, 27 actuators, and an internal network were gathered. The data from sensors and actuators were measured every second, along with about 350 samples of network traffic data. The SWaT dataset was collected over 11 days of continuous operation, of which the data from the first 7 days recorded only normal operations, whereas the remaining 4 days included attack data. In addition, each record in the dataset was labeled as representing either a normal or an attack condition. In total, 36 attacks were launched on the SWaT testbed, which were classified into four types, as shown below:

1.  Single Stage Single Point (SSSP): an attack on a single sensor/actuator value in any single stage (e.g., Attack 1 on LIT101 in P1 stage);
2.  Single Stage Multi-Point (SSMP): an attack on multiple sensor/actuator values in any single stage (e.g., Attack 16 on MV101 and LIT101 in P1 stage);
3.  Multi-Stage Single Point (MSSP): an attack on a single sensor/actuator value over multiple stages (e.g., Attack 21 on P101 in P1 stage and LIT301 in P3 stage);
4.  Multi-Stage Multi-Point (MSMP): an attack on multiple sensor/actuator values over multiple stages (e.g., Attack 17 on UV401 in P4 stage and AIT502, P501 in P5 stage).

For SSSP, SSMP, MSSP, and MSMP, there were 26, 4, 2, and 4 attacks, respectively.

In general, an ICS is maintained by various process control loops (PCLs). Figure 2 presents a simple diagram of the SWaT ICS testbed, including the six processes, P1 through P6, where each process is controlled by a corresponding programmable logic controller (PLC) that is connected to a system administrator through a communication network. First, the system administrator sets the target values based on their ICS operation plan using a human–machine interface (HMI). For example, they can set a target for the amount of water treated per unit of time. Then, these target values are transferred to a system controller, such as a PLC or a distributed control system (DCS), through a communication network between the HMI and PLCs/DCSs. Second, the controller inputs calculated values for actuators in the ICS that are based on the target values. The actuators are set by received values. Then, the ICS operates the system and measures the current values using embedded sensors. Third, these measured sensor values are transferred to the system controller and the controller calculates the actuator values again to achieve the target values. In

the second and third steps, the values of the actuators and sensors are transferred through a communication network between the PLCs/DCSs and the actuators/sensors (see the blue and red lines in Figure 2.) Finally, the recalculated actuator values are transferred to the ICS and this process loop is iterated continuously during the operation of the ICS. The SWaT dataset includes attacks targeting both sensors and actuators. As an example of a sensor attack, the dataset includes an incident in which attacks set the FIT-401 sensor value to 0, which stops the ultraviolet dechlorination process. In an actuator attack recorded in the dataset, attackers shut off the P-302 pump, which in turn stops the water supply to the T-401 water tank.

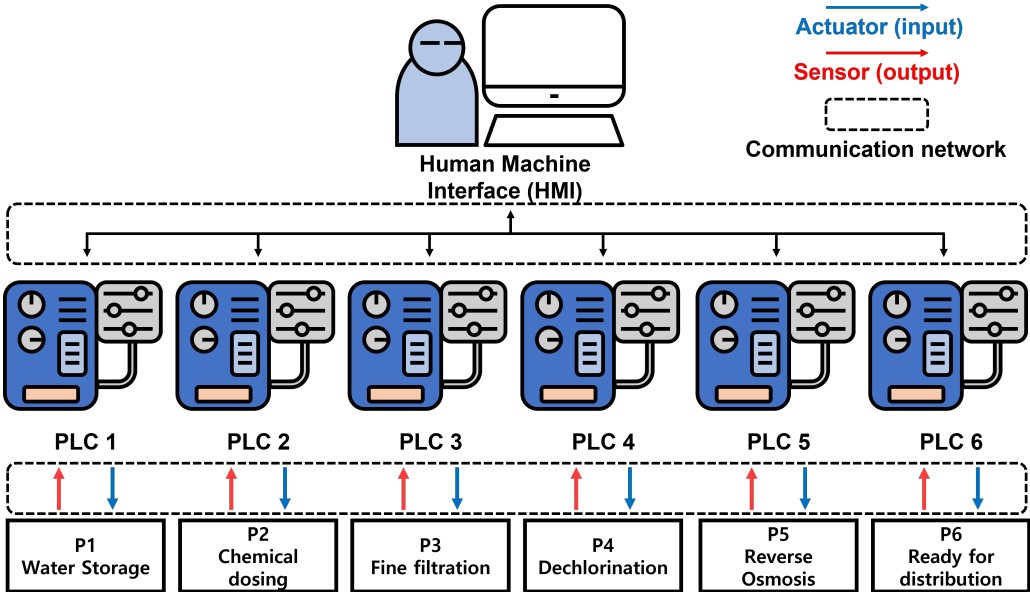

**Figure 2.** An example of the feedback loops in an ICS (the SWaT testbed example).

As the SWaT dataset is publicly available, many researchers have used it to perform case studies for methods involving ICS technologies. In 2017, Goh et al. used this dataset to conduct research on anomaly detection based on a LSTM learning architecture [31]. Although they showed that LSTM is useful for detecting anomalies, they only focused on data collected from stage P1 and omitted the remaining five stages. Lin et al. proposed a method in 2018 that combined timed learning automata and Bayesian network inference to perform anomaly detection and then evaluated it using the SWaT dataset [32]. Their method, TABOR, showed better performance than deep neural network (DNN) and SVM learning models. In 2020, Mieden et al. proposed a method to detect anomalies that was tested on the SWaT dataset using only the network traffic data [33]. They used 16 network features out of 19 in total and implemented an LSTM as an anomaly detection model. In this study, following on from many previous works [11,31,32,34,35], we used the time series data of the sensors and actuators.

### 2.4. Composite Autoencoder

An autoencoder (AE) is a representative unsupervised learning architecture. Typically, this type of neural network encodes input data to a lower-dimensional vector (i.e., less latent variables) and decodes latent variables to restore the input data. Due to this structure, AE networks are trained to generate latent variables that represent the features of input data well. With this property, AEs have been typically used to perform dimensionality reduction [36,37]. Although they typically perform well at decoding data that are similar to training data, AEs tend to produce poor reconstruction outputs when the input data differ significantly from the training data. Therefore, anomaly detection models are generally trained only on normal data, assuming that normal and attack data are different. Then, the agent passes newly received data to the model to determine the extent to which it is

anomalous. Detection models are designed to output a small test error for normal data and to output a large test error otherwise. By setting an appropriate threshold to distinguish normal input from attack input, the agent can effectively detect anomalies. As a result of this functionality, AEs have been considered in many studies on anomaly detection. In 2017, Aminanto et al. proposed a fully unsupervised method to detect impersonation attacks in a Wi-Fi network [8]. Their proposed method was composed of a stacked AE that was designed to extract features and a k-means clustering method that was used to detect malicious data. In 2020, Park et al. used an AE to identify attacks targeting an unstaffed aerial vehicle [9]. The authors trained an AE to reconstruct an original input vector as an output and they recognized DoS and GPS spoofing attacks by measuring reconstruction loss.

For ICS security, An et al. proposed an anomaly detection method based on variational AEs in 2015 [10]. The authors evaluated their method using the MNIST [38] and KDD Cup 1999 network intrusion dataset [26]. Similarly, in [11], Wang et al. proposed a composite autoencoder (CAE) model for anomaly detection in ICSs. This model was evaluated using the SWaT dataset [25]. In addition, Chang et al. proposed an anomaly detection method combining k-means clustering and a convolutional AE [12]. Their method was evaluated using two ICS log datasets based on data representing a gas pipeline [29] and a water storage tank [28]. In 2020, Jones et al. used both signature-based and behavior-based methodologies independently to monitor and detect anomalous traffic between an aggregator and a single photovoltaic inverter caused by network-based cyberattacks [39]. In this research, the authors considered an autoencoder as a behavior-based detection method.

In this study, we adopted the CAE model proposed in [11], in which an LSTM was used as a building block. The whole model is shown in Figure 3. A CAE is a neural network incorporating a single encoder and two decoders. This model can overcome the shortcomings that occur when running a reconstruction or prediction model alone [40]. The first decoder of a CAE is designed to learn to reconstruct the input data, whereas the second decoder learns to predict the data for the next step. This combination of overlapping time steps renders a CAE particularly useful for anomaly detection for time series data, such as the SWaT dataset.

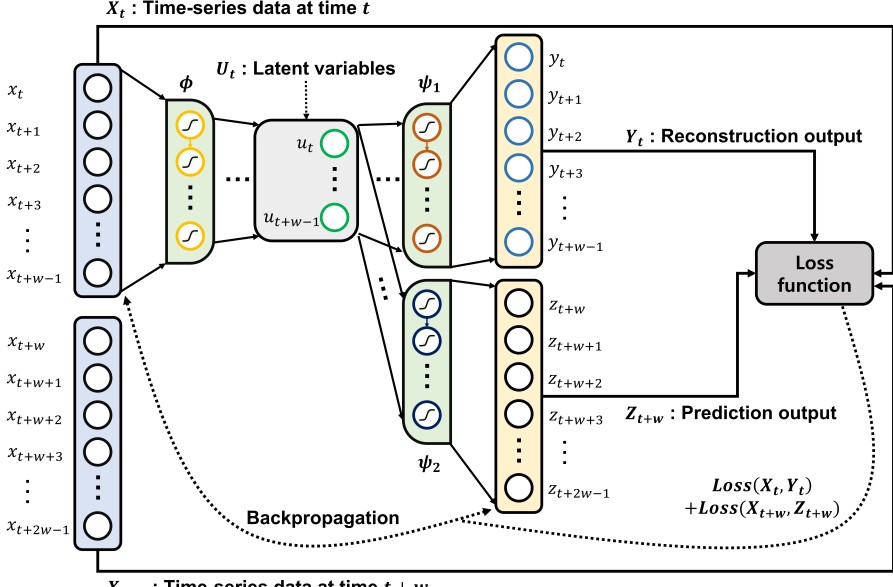

**Figure 3.** An example of a composite autoencoder.

For example, let $X_t = (x_t, x_{t+1}, \ldots, x_{t+w-1}) \in \mathbb{R}^{n \times w}$ be an input data point, which is the time series data at time $t$. Each $x_i$ is composed of $n$ features and $X_t$ is composed of $w$

consecutive $x_i$'s, where $w$ is the time window size. $X_{t+w}$ is the next time step data following $X_t$. By modifying the equation for the general autoencoder in [41], the three components of a CAE can be defined as follows:

$$\phi : \mathbb{R}^{n \times w} \to \mathbb{R}^{l \times w},$$
$$\psi_1 : \mathbb{R}^{l \times w} \to \mathbb{R}^{n \times w},$$
$$\psi_2 : \mathbb{R}^{l \times w} \to \mathbb{R}^{n \times w},$$

where $\phi$ is an encoder, $\psi_1$ is a decoder for reconstruction, and $\psi_2$ is a decoder for prediction. For an input data point $X_t$, the CAE encodes $X_t$ to the latent variables $U_t = (u_t, u_{t+1}, \ldots, u_{t+w-1}) = \phi(X_t)$, where each $u_i$ is composed of $l$ features. Next, it decodes $U_t$, producing two outputs: $Y_t = (y_t, y_{t+1}, \ldots, y_{t+w-1}) = \psi_1 \circ \phi(X_t)$ for reconstruction and $Z_{t+w} = (z_{t+w}, z_{t+w+1}, \ldots, z_{t+2w-1}) = \psi_2 \circ \phi(X_t)$ for prediction. Then, $\phi$, $\psi_1$, and $\psi_2$ are optimized to satisfy:

$$\underset{\phi, \psi_1, \psi_2}{\operatorname{argmin}} (Loss(X_t, \psi_1 \circ \phi(X_t)) + Loss(X_{t+w}, \psi_2 \circ \phi(X_t))),$$

where *Loss* is the loss function, which is explained in detail in Section 4.

## 3. Proposed Method

In this section, we detail the proposed hybrid anomaly detection method in which anomalous data are first filtered by a statistical analysis unit before a CAE is used to reduce false negatives. Figure 4 presents the overall flow of the proposed hybrid anomaly detection method when a sensor in Process 1 of the water treatment ICS is attacked. The water treatment system treats water and collects the values transmitted by the sensors and actuators. Then, a security gateway gathers the values generated by the system and sends them to a cloud server, i.e., a water treatment system monitoring and control system, which then reports the system data via the industrial supervisory control and data acquisition (SCADA) network. The cloud server stores the system data gathered from the water treatment system and performs analytics based on the measured amount of treated water to improve the system plan. Sophisticated attackers in possession of extensive knowledge of these systems, e.g., advanced persistent threats (APT) [42] and insider threats [43], may persist in attempting to attack the ICS system, aiming to compromise its security and gain the ability to damage or destroy the system or to forge its data. To take these advanced attackers into account, we assume that an attacker has already gained the ability to control the sensors and actuators. The detection process is performed in the security gateway. After receiving the data values from the water treatment system, the gateway determines whether the state of the system is interpreted as normal using the proposed hybrid detection method.

When the gateway receives the data from the system, the anomaly detection process begins. In the process, the data are classified by a signature-based anomaly detection method in which a statistical analysis is performed. When the data are labeled as normal, they are passed to a behavior-based anomaly detection model; we adopted the CAE model proposed in [11] as the behavior-based anomaly detection model. After the detection processes are performed, the security gateway labels the data. When the data sample is labeled as normal, it is sent to the cloud server; otherwise, the security gateway reports that an attack has been detected to the system administrator.

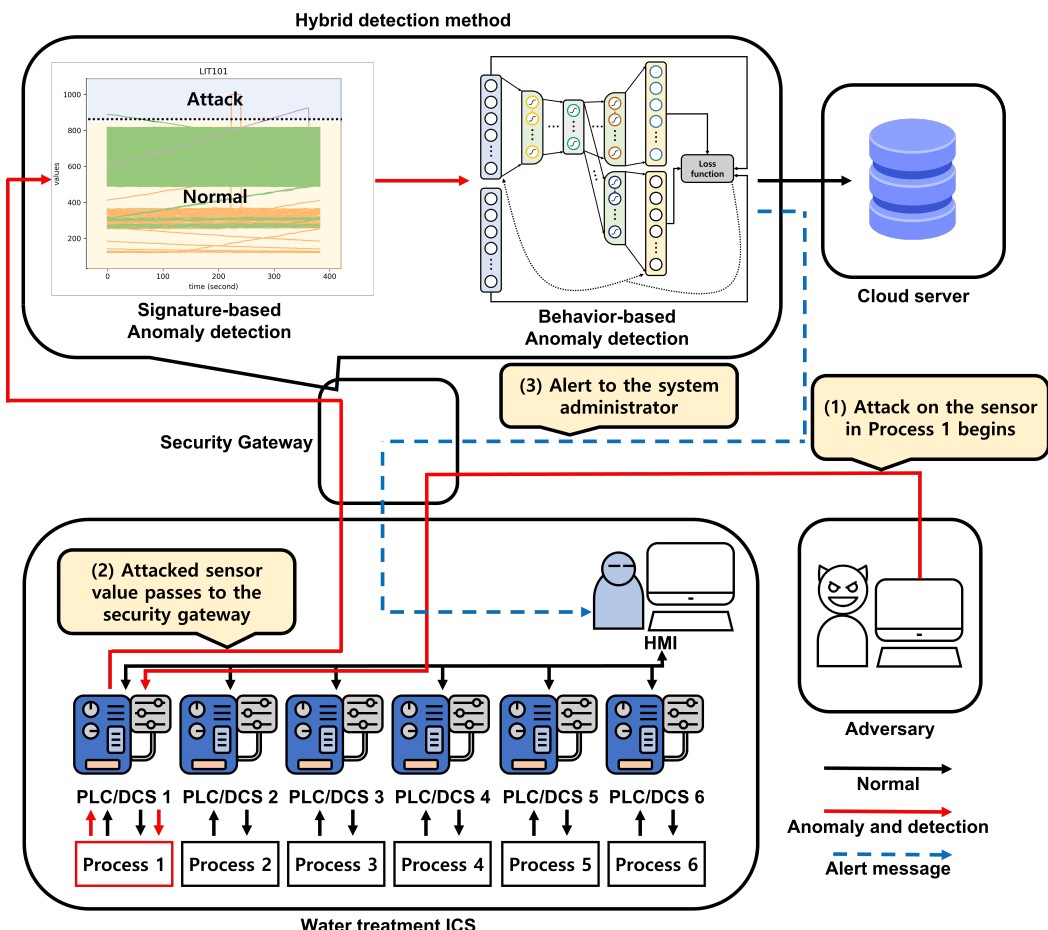

**Figure 4.** The overall flow of the proposed hybrid anomaly detection method.

In the signature-based anomaly detection process, the input data are filtered based on statistical features. To filter malicious data, we use the standard deviation of the normal data, comprising a time series sequence $(d_1, d_2, \dots)$ where each $d_i$ is sampled at one-second intervals. Each $d_i$ is composed of $n$ elements, $v_{i,1}, v_{i,2}, \dots, v_{i,n}$, where $n$ is the number of sensors and actuators in the dataset. Let $D = (d_t, d_{t+1}, \dots, d_{t+w-1})$ be the length $w$ of a segment of the data beginning at a specific time $t$, where $w$ is the window size. In the training stage, $n$ standard deviations $S_1^D, S_2^D, \dots, S_n^D$ are computed for each possible segment $D$ of normal data, where $S_j^D = \sigma(v_{t,j}, v_{t+1,j}, \dots, v_{t+w-1,j})$ is the standard deviation of $w$ consecutive values $v_{t,j}, v_{t+1,j}, \dots, v_{t+w-1,j}$ for the $j$-th sensor or actuator. Then, $Min_{S_j} = \min_D S_j^D$ and $Max_{S_j} = \max_D S_j^D$ are computed over all possible segments $D$. For example, we can consider $D = (d_1, d_2, \dots, d_w), (d_6, d_7, \dots, d_{w+5}), (d_{11}, d_{12}, \dots, d_{w+10}), \dots$ so that the windows overlap every 5 s. The dataset is explained in detail in the next section. $Min_{S_j}$ and $Max_{S_j}$ are the lower and upper bounds of the standard deviation of the $w$ consecutively reported normal values for sensor/actuator $j$, respectively.

When a test input segment composed of $w$ consecutive samples $X_i = (d_i, d_{i+1}, \dots, d_{i+w-1})$ is processed, the method computes $P_1, P_2, \dots, P_n$, where $P_j$ is a boolean predicate such that $P_j = True$ when $Min_{S_j} \leq \sigma(v_{i,j}, v_{i+1,j}, \dots, v_{i+w-1,j}) \leq Max_{S_j}$, otherwise $P_j = False$. The input is labeled as normal when $P_1 \cap P_2 \cap \dots \cap P_n = True$, i.e., the standard deviations of all sensors/actuators are within the normal range; otherwise, it is filtered as an attack. When the input is labeled as normal, it is forwarded to the CAE for the next intrusion detection process to further reduce false negative rates. Then, when the output error of the CAE is greater than a threshold, it is labeled as an attack; otherwise, it is labeled as normal.

## 4. Implementation and Validation

In this section, we validate the performance and evaluate the execution time of the proposed method. We also compare them to those of the CAE-only method. The experimental environment used to implement the proposed method primarily comprised a machine with an Intel Xeon Gold 6242 CPU @ 2.8 GHz with 256 GB RAM and an NVIDIA TITAN RTX GPU with 24 GB of GDDR6 memory, which was used to train and validate the proposed model. The CAE model was implemented on the Keras library version 2.4.3 and scikit-learn version 0.22.2.post1.

Before training the model, the data preprocessing was performed as follows. First, as in [11], we removed unstable data columns with markedly different distributions between the training and testing datasets, including AIT201, P201, FIT601, P601, P602, and P603. Second, the training dataset was scaled to the interval $(0, 1)$ using the MinMaxScaler class of the scikit-learn library and the testing dataset was also scaled using the minimal and maximal values of each column obtained from the training dataset. Third, we divided the dataset using a sliding window. For the training dataset, we overlapped the windows to maximize the performance as presented in Figure 5, but the test dataset was simply divided without an overlap. In this experiment, we set the window size $w$ to 120 with an overlap size of 115, following the setup in [11].

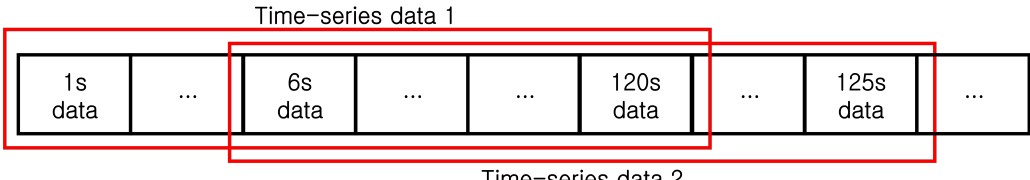

**Figure 5.** The division of the training dataset using a sliding window.

We implemented the CAE model based on an LSTM building block, which is considered to be a suitable architecture for time series data. Although we referred to the model from [11], we implemented our own model that was as similar as possible to that described in [11] because the code of [11] was not publicly available. With a trial and error approach, the hyperparameters used for the model were selected as follows:

- #Neurons in encoder layers 1 and 2: 64 and 32, respectively;
- #Neurons in latent variables: either 8 or 16 (we provided two versions.);
- #Neurons in decoder layers 1 and 2: 32 and 64, respectively;
- Activation function: hyperbolic tangent;
- Loss function: mean squared error;
- Optimizer: Adam optimizer.

A CAE model outputs two decoding results for a single input segment: one for reconstruction and the other for prediction. Therefore, we calculated two errors from the results. We let $X_t$ be a test input segment starting at time $t$, i.e., $X_t = (d_t, d_{t+1}, \ldots, d_{t+w-1})$, where each $d_i$ was composed of $n$ elements, $v_{i,1}, v_{i,2}, \ldots, v_{i,n}$. For the input segment $X_t$, the CAE outputted a reconstruction result $Y_t = (d'_t, d'_{t+1}, \ldots, d'_{t+w-1})$ and a prediction result $Z_t = (d''_t, d''_{t+1}, \ldots, d''_{t+w-1})$.

Then, we could calculate two errors $e_t^{recon}$ and $e_t^{pred}$ at time $t$ from the outputs of the CAE and the overall error $E_t$ as follows:

$$e_t^{recon} = (|v_{t,1} - v'_{t,1}|, |v_{t,2} - v'_{t,2}|, \ldots, |v_{t,n} - v'_{t,n}|),$$

$$e_t^{pred} = (|v_{t,1} - v''_{t,1}|, |v_{t,2} - v''_{t,2}|, \ldots, |v_{t,n} - v''_{t,n}|),$$

$$E_t = e_t^{recon} + e_t^{pred}. \tag{1}$$

We note that we previously set the time windows for the reconstruction and prediction errors to different periods in the preliminary version of this work [13]. That is, the overall error $E'_t$ used in [13] was calculated by:

$$E'_t = e^{recon}_t + e^{pred}_{t+w}. \tag{2}$$

To set the threshold for attack detection, we applied an exponentially weighted moving average method (EWMA) and a power technique according to [11]. Then, the smoothed error $SE_t$ and $p$-power error $PE_t$ were calculated as follows:

$$SE_t = \begin{cases} 0, & \text{if } t = 0, \\ \alpha E_t + (1-\alpha)SE_{t-1}, & \text{otherwise,} \end{cases}$$

$$\alpha = 1 - exp^{\frac{ln(0.5)}{H}},$$

$$PE_t = \frac{1}{n}\sum_{j=1}^{n}|SE_t^j|^p,$$

where $SE_t^j$ is the $j$-th element in the $n$-dimensional vector $SE_t$ and we set the half-life period $H = w$ and the parameter of $p$-powered error as $p = 4$, according to [11]. When the calculated $p$-powered error $PE_t$ was greater than or equal to the threshold $V_{thre}$, the data at time $t$ were classified as anomalous, where the $V_{thre}$ was the maximal $PE_t$ determined in the training stage. That is, the anomaly flag $A_t$ was defined by:

$$A_t = \begin{cases} 1, & PE_t \geq V_{thre}, \\ 0, & \text{otherwise.} \end{cases}$$

As the input segment $X_t$ was the time series data for $w$ seconds, we obtained $w$ labels for $X_t$. Finally, when there was at least one attack label in the input, it was classified as an attack.

We adopted True Positive (TP), True Negative (TN), False Positive (FP), and False Negative (FN) rates as performance metrics, as presented in Table 1. We also adopted the commonly used representative metrics of precision $= \frac{TP}{TP+FP}$, recall $= \frac{TP}{TP+FN}$, and F1-score $= \frac{2\times\text{Precision}\times\text{Recall}}{\text{Precision}+\text{Recall}}$.

**Table 1.** The confusion matrix.

| True Class | Classified as Anomaly | Classified as Normal |
|---|---|---|
| Anomaly | True Positive (TP) | False Negative (FN) |
| Normal | False Positive (FP) | True Negative (TN) |

Figure 6 shows the confusion matrices for the hybrid methods with 8 and 16 dimensions of latent variables. The precision, recall, F1-score, and accuracy were computed based on these matrices. Table 2 presents the experimental results of the performances of the CAE-only method, the hybrid method in [13] with the error $E'_t$ (Equation (2)), and the hybrid method with the error $E_t$ (Equation (1)). The rows labeled "CAE-only method" demonstrate the results when the anomaly detection was performed using only a CAE. As mentioned above, the code for the CAE-only model in [11] was not available. Therefore, we tried to implement the CAE model as closely as possible to the description in [11]. Then, we compared the performance of this implementation to that of the proposed approach. We validated the performance in comparison to two CAE models with 8 and 16 different dimensions of latent variables. In addition, we applied the proposed hybrid method to each model and measured their performance. According to the results, the recall and F1-score for the hybrid method 1 improved by up to 0.078 and 0.025, respectively, and the precision

decreased by up to 0.067 compared to those of the CAE-only method. For example, when the dimension of latent variables was 8, the increase in the recall was $0.856 - 0.778 = 0.078$. On average, the recall and F1-score increased by 0.076 and 0.016, respectively, while the precision decreased by about 0.048.

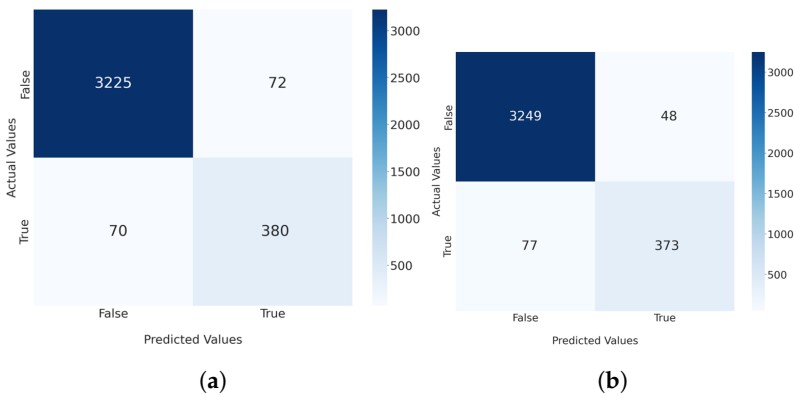

(**a**)　　　　　　　　　　　　　(**b**)

**Figure 6.** The confusion matrices for our hybrid method. (**a**) The confusion matrix for the case with 8 latent variables. (**b**) The confusion matrix for the case with 16 latent variables.

**Table 2.** The measured performances of the various methods.

| #Neurons of Layers | Anomaly Detection Method | Precision | Recall | F1-Score |
|---|---|---|---|---|
| (64, 32), 8, (32, 64) | CAE-only method | 0.833 | 0.778 | 0.805 |
| | Hybrid method 1 [13] | 0.805 | 0.856 | 0.830 |
| | Hybrid method 2 | 0.841 | 0.844 | 0.843 |
| (64, 32), 16, (32, 64) | CAE-only method | 0.879 | 0.762 | 0.817 |
| | Hybrid method 1 [13] | 0.812 | 0.836 | 0.824 |
| | Hybrid method 2 | 0.886 | 0.829 | 0.856 |

In contrast, when the hybrid method 2 was applied, all of the precision, recall, and F1-score results improved by up to 0.008, 0.067, and 0.039, respectively. On average, the precision, recall, and F1-score increased by approximately 0.008, 0.067, and 0.039, respectively. In addition, the results showed that the accuracy was 0.962 and 0.967 when the dimensions of latent variables were 8 and 16, respectively. As demonstrated in the results, it can be clearly observed that the hybrid method detected anomalies that might not have been detected by the CAE-only approach. However, for hybrid method 1, a slightly higher number of false alarms were observed compared to the CAE-only method. It is conjectured that the precision decreased because some normal data were filtered as an attack when the filtering of hybrid method 1 was applied. These experimental results demonstrate that the model with 8 dimensions of latent variables exhibited a slightly better performance in terms of F1-score for hybrid method 1, whereas the model with 16 dimensions of latent variables showed a better performance for hybrid method 2. In summary, the proposed filtering method improved the overall anomaly detection performance and the proposed hybrid method performed better than the CAE-only method.

Comparing the two hybrid methods, hybrid method 2 showed better precision and F1-score by up to 0.074 and 0.032, respectively, whereas the recall decreased by up to 0.012. On average, the precision and F1-score increased by 0.055 and 0.023, respectively, while the recall decreased by 0.01. According to the comparison between the two methods, an ICS administrator could use hybrid method 1 to detect more anomalies, which allowed more false alarms. Alternatively, they could also apply hybrid method 2 for a more balanced and reliable detection system that generated fewer false alarms.

Figure 7 depicts the receiver operating characteristic (ROC) curves of the proposed hybrid detection method. The curves were derived from the observed results by setting

$V_{thre}$ from the minimal to maximal values, where $V_{thre}$ is the threshold used to determine whether the data sample is an anomaly or not. The figure demonstrates that the proposed method guaranteed high detection performance with area under curves (AUCs) of 0.955 and 0.952 for the methods with 8 and 16 dimensions of latent variables, respectively.

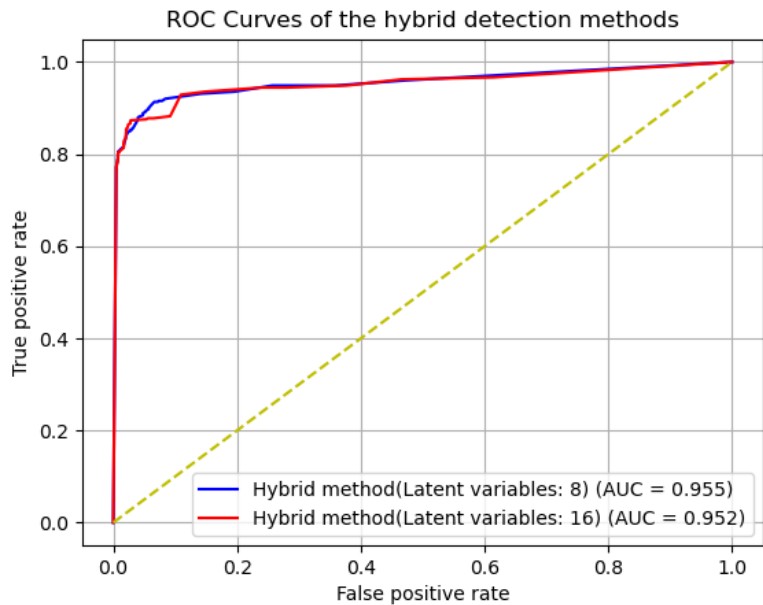

**Figure 7.** The ROC curves for the hybrid method with 8 and 16 dimensions of latent variables.

To verify the performance improvement of the proposed method, we compared its performance to the experimental results for other machine learning models presented in [32]. Table 3 presents the comparison of the performance of the proposed method and those of DNN, SVM, and TABOR in [32]. The precision of the proposed method was lower than those of DNN and SVM, but higher than that of TABOR. However, the proposed method showed the best recall and F1-score of all of the methods.

**Table 3.** A performance comparison between the proposed method and DNN, SVM, and TABOR in [32].

| Method | Precision | Recall | F1-Score |
|---|---|---|---|
| Hybrid Method 2 (Latent variables: 16) | 0.886 | **0.829** | **0.856** |
| Deep Neural Network (DNN) | **0.983** | 0.678 | 0.803 |
| Support Vector Machine (SVM) | 0.925 | 0.699 | 0.796 |
| TABOR | 0.862 | 0.788 | 0.823 |

We conducted an additional experiment to examine the influence of standard deviation bounds in the first stage (i.e., signature-based anomaly detection) on the overall performance. In Section 3, we set the lower and upper bounds of the standard deviation for sensor/actuator $j$ as simply $Min_{S_j}$ and $Max_{S_j}$, respectively. In the additional experiment, we evaluated the performance by changing the lower and upper bounds. After trying various combinations, we were able to obtain a slightly better performance when we used larger upper bounds than $Max_{S_j}$ with the original lower bound $Min_{S_j}$ unchanged. Table 4 shows the experimental results in detail, where the upper bound was set to $Max_{S_j} \times \beta$ and $\beta$ was 1.0, 1.2 or 1.5. According to the experimental results, the precision and F1-score increased by up to 0.006 and 0.003, respectively. This indicates that a generous upper bound was able to slightly reduce false alarms.

**Table 4.** The performance of larger upper bound.

| #Neurons of Layers | Multiplier $\beta$ | Precision | Recall | F1-Score |
|---|---|---|---|---|
| (64, 32),  8, (32, 64) | 1.0 (Baseline) | 0.841 | 0.844 | 0.843 |
| | 1.2 | 0.844 | 0.844 | 0.844 |
| | 1.5 | 0.844 | 0.844 | 0.844 |
| (64, 32), 16, (32, 64) | 1.0 (Baseline) | 0.886 | 0.829 | 0.856 |
| | 1.2 | 0.892 | 0.829 | 0.859 |
| | 1.5 | 0.890 | 0.829 | 0.858 |

Finally, we analyzed the execution times of the CAE-only and hybrid methods. Table 5 presents the measured average times to classify one input segment for each method. Initially, it may be expected that the hybrid detection method would take longer as it includes a filtering method before the CAE. Interestingly, however, the hybrid detection method was faster than the CAE-only method according to our results. The execution time of the hybrid method with 8 latent variables was 141.812 ms, whereas the CAE-only method required 150.082 ms on average. For the other model, the execution times of the hybrid method and the CAE-only method were 137.163 ms and 149.143 ms, respectively. Consequently, the hybrid model reduced the time required by approximately 5.51% and 8.03%, respectively.

**Table 5.** The measured times for anomaly detection.

| #Neurons of Layers | Anomaly Detection Method | Average Time (ms) |
|---|---|---|
| (64, 32),  8, (32, 64) | CAE-only method | 150.082 |
| | Hybrid method | 141.812 |
| (64, 32), 16, (32, 64) | CAE-only method | 149.143 |
| | Hybrid method | 137.163 |

To consider why the hybrid method was able to reduce the execution time, let us recall that an anomaly could be detected in two different ways for the hybrid method. The first is the case in which the filtering method classifies the input segment as an anomaly. In this case, the hybrid detection method can skip the second stage, i.e., the behavior-based detection process. The second case corresponds to the situation where the filtering method classifies the input as normal, but the CAE in the second stage classifies the input segment as an anomaly. Let $N$, $n$, $T_S$, and $T_B$ be the total number of input segments to be analyzed, the number of segments classified as anomalous, the execution time of the filtering stage, and the execution time of the CAE, respectively. Then, we can estimate the execution times of the CAE-only and hybrid methods using the following equations:

$$\text{CAE-only method} = T_B \times N.$$
$$\text{Hybrid method} = T_S \times n + (T_S + T_B)(N - n)$$
$$= T_S \times N + T_B \times N - T_B \times n.$$

That is, the hybrid detection method performs anomaly detection faster than the CAE-only method when $T_S \times N < T_B \times n$, i.e., $n/N > T_S/T_B$. For example, according to the experimental results with 8 dimensions, the hybrid method took 0.215 ms on average when it detected an anomaly in the signature-based anomaly detection process, i.e., $T_S = 0.215$ ms. According to Table 5, $T_B = 150.082$ in this setting. When both stages were performed by the hybrid method, the computation had a duration of 156.355 ms according to the experimental result. This was slightly larger than $T_S + T_B$, which could be due to the additional overheads

in the experiment. In this case, the hybrid detection method would have performed faster than the CAE-only method if $\frac{n}{N} > \frac{0.215}{156.355} = 0.00138$, which was the case for the SWaT dataset. By analyzing the proposed method in terms of the operation time, we showed that the hybrid detection method could reduce detection overheads by decreasing the number of CAE operations.

In summary, the experimental results show that the proposed hybrid detection method performed anomaly detection better than the CAE-only method and reduced the detection overheads by decreasing the number of CAE operations. Based on these results, we expect that existing anomaly detection methods that only use behavior-based detection can be improved by applying the proposed hybrid approach and that the proposed method can be considered a suitable option for resource-constrained environments.

## 5. Discussion and Limitations

In this paper, we proposed a hybrid anomaly detection method that combines signature-based and behavior-based methods. The signature-based method uses statistical filtering based on the standard deviation of time series data and the behavior-based method uses a CAE. According to our experimental results, the proposed method outperformed the CAE-only method and other machine learning-based methods, such as DNN, SVM, and TABOR.

Although the proposed method showed excellent performance in detecting anomalies, a few interesting research topics need to be addressed in future work. First, the proposed method suffers from the same limitation as [31] in that it only detects the fact that an anomaly occurred but does not determine which sensors or actuators were attacked. Therefore, we will pursue research to provide more precise detection in future work.

Second, we only used conventional metrics, such as precision, recall, F1-score, and accuracy, to evaluate the performance in this paper. However, several alternative metrics may be more suitable for time series data, such as the measured values of the sensors and actuators. For example, Tatbul et al. proposed range-based precision and recall that consider partial overlaps between the real and predicted ranges and their relative positions [44]. Hwang et al. also considered partially overlapped ranges and proposed time series aware precision (TaP) and time series aware recall (TaR) [45]. They also considered the ambiguous period in which the data were still affected by the precedent anomaly even after the attack ceased. Therefore, the evaluation of the proposed method using these metrics would be an interesting research topic.

Third, we used a trial and error approach to search for the optimal dimension of the latent variables and the optimal activation function for the CAE because they were not explicitly described in the original proposal of CAE-based anomaly detection [11]. However, extensive research has been conducted to systematically find optimal neural network architectures and hyperparameters. In 2018, Kaspersky proposed genetic algorithms (GAs) to find the best neural network architecture for anomaly detection [46]. In 2019, Jin et al. proposed an efficient neural architecture search framework [47]. They used Bayesian optimization to guide the search space and published Auto-Keras, an open-source AutoML system that is based on this method. In 2021, Alharbi et al. proposed a neural network-based optimized method called the local global best bat algorithm for neural network (LGBA-NN) [48]. The method selects both feature subsets and hyperparameters to efficiently detect botnets and was tested using an N-BaIoT dataset that included extensive real traffic data. It would be a promising research direction to apply these neural network architecture search techniques to improve the proposed hybrid detection method.

## 6. Conclusions

In this study, we proposed a hybrid anomaly detection method combining signature-based and behavior-based methods for a real-time control system using statistical filtering and a CAE, respectively. To validate the performance of the proposed method, we conducted experiments using the SWaT dataset for a real water treatment system. The results

show that the proposed hybrid detection method outperformed the CAE-only method in terms of detection accuracy, as measured by the precision, recall, and F1-score. According to the experimental results, the proposed method improved the precision, recall, and F1-score by up to 0.008, 0.067, and 0.039, respectively, compared to an autoencoder-only approach. It also reduced the time required to execute the detection process by up to 8.03% compared to the autoencoder-only method. As the proposed method is universal, it can be applied to any other ICSs, such as electric power grids and smart factories. We expect that existing anomaly detection methods that only use behavior-based detection can be improved by applying the proposed hybrid detection approach and that the proposed method can be considered as a suitable option for resource-constrained environments.

**Author Contributions:** Conceptualization, M.-K.L.; funding acquisition, M.-K.L.; investigation, H.-Y.K. and T.K.; project administration, M.-K.L.; supervision, M.-K.L.; validation, H.-Y.K.; writing—original draft, H.-Y.K.; writing—review and editing, T.K. and M.-K.L. All authors have read and agreed to the published version of the manuscript.

**Funding:** This work was supported in part by the MSIT, Korea, under the High-Potential Individuals Global Training Program (grant number: 2020-0-01540) and supervised by the IITP, also in part by the IITP grant funded by the Korean government (MSIT) (2020-0-01389, Artificial Intelligence Convergence Research Center (Inha University)), and in part by the Inha University Research Grant.

**Data Availability Statement:** Restrictions apply to the availability of these data. Data were obtained from iTrust and are available at https://itrust.sutd.edu.sg/itrust-labs_datasets/dataset_info/ (accessed on 23 February 2022) with the permission of iTrust, Centre for Research in Cyber Security, Singapore University of Technology and Design.

**Acknowledgments:** This work was conducted in part while Hee-Yong Kwon was visiting Texas A&M University Kingsville, Kingsville, TX, USA.

**Conflicts of Interest:** The authors declare no conflict of interest.

**Abbreviations**

The following abbreviations are used in this manuscript:

| | |
|---|---|
| AE | Autoencoder |
| APT | Advanced Persistent Threats |
| AUC | Area Under Curve |
| CAE | Composite Autoencoder |
| CPS | Cyber-Physical System |
| DCS | Distributed Control System |
| DNN | Deep Neural Network |
| DoS | Denial of Service |
| EWMA | Exponentially Weighted Moving Average |
| FN | False Negative |
| FP | False Positive |
| GA | Genetic Algorithms |
| HIDS | Host-based Intrusion Detection System |
| HIL | Hardware-in-the-Loop |
| HMI | Human–Machine Interface |
| ICS | Industrial Control System |
| IDS | Intrusion Detection system |
| IoT | Internet of Things |
| LAN | Local Area Network |
| LSTM | Long Short-Term Memory |
| MSMP | Multi-Stage Multi-Point |
| MSSP | Multi-Stage Single Point |
| NIDS | Network-based Intrusion Detection System |

| PCL | Process Control Loop |
| PLC | Programmable Logic Controller |
| ROC | Receiver Operating Characteristic |
| SCADA | Supervisory Control and Data Acquisition |
| SSMP | Single Stage Multi-Point |
| SSSP | Single Stage Single Point |
| SVM | Support Vector Machine |
| SWaT | Secure Water Treatment |
| TCP | Transmission Control Protocol |
| TN | True Negative |
| TP | True Positive |

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
