# Peer review of "Advanced Intrusion Detection Combining Signature-Based and Behavior-Based Detection Methods†"

_electronics, doi:10.3390/electronics11060867_

Round 1

Reviewer 1 Report

The paper proposes an approach for network intrusion detection that is based on the signature-based and behavior-based detection methods. The results of experiments on the SWaT dataset are presented.

Comments:

  1. Summarize the main findings in the abstract, including a summary of the main experimental results.
  2. Explicitly state the novelty of this study at the end of the introduction section.
  3. The overview of related works is very narrow and focuses only on the works using the Swat dataset, and a few using autoencoders. However, the domain of network intrusion detection is much larger. I suggest to expand this part of the paper by discussing state-of-the-art of network intrusion detection in general. For example, you can discuss “A novel approach for network intrusion detection using multistage deep learning image recognition”, and “Botnet Attack Detection Using Local Global Best Bat Algorithm for Industrial Internet of Things”, among others.
  4. There are many network intrusion datasets available. Why did you select the SWaT dataset for your study? Did you consider any alternatives? Add a short discussion on benchmark network intrusion datasets.
  5. What is the meaning of Figure 2? The figure is not related to network intrusion.
  6. Figure 4 is too informal and lacks of technical detail. I suggest to include a more formal workflow diagram demonstrating the working of your method.
  7. How was window size and overlap size selected (Line 260)? Did you perform any ablation study?
  8. Why did you use the trial-and-error approach for the selection of hyperparameter values rather than existing hyperparameter optimization frameworks such as Autokeras?
  9. Present and discuss more experimental results such as confusion matrix, ROC plot, AUC value.
  10. Add a discussion on the limitations of the proposed methodology.
  11. In the Conclusions section, use the main numerical findings of your study to support your claims.

Reviewer 2 Report

This paper is deploying a composite autoencoder in order to perform anomaly detection through signature-based as well as behaviour-based identification methods. This paper needs some improvement before consideration for publication as follows. 

  1. The authors need to highlight the novelty and contributions of this work. It seems the authors have used a publicly available dataset and an existing CAE technique to detect anomalies. The significance of the paper cannot be highlighted without properly highlighting the novelty and contributions of this paper. 
  2. The section 2.2 "Composite Autoencoder" needs further theoretical explanation about how the CAE generates two different outputs. Without proper theoretical and mathematical explanations, the working of CAE cannot be followed. 
  3. The performance of the CAE needs to be evaluated against other ML models such as Neural network, deep neural network, KNN etc., including conventional AEs, to prove efficiency of CAE. 
  4. The system model needs further clarification and detailed explanation about in which node the proposed system will be implemented. The Figure 4 needs to be modified accordingly so that it can reflect the whole system model.
  5. What is the difference between server and victim systems? Need more explanations
  6. Accuracy is a commonly adopted performance measure in ML-based detection applications. Why is it not included in this paper?
  7. There are typos and grammatical mistakes throughout the paper that needs correction. For instance, line 88 has "?" sign in the citation brackets "[]"

Round 2

Reviewer 1 Report

The authors have addressed all my concerns and revised the article accordingly. I have no further comments and recommend to accept.

Reviewer 2 Report

This paper has been improve significantly and the authors have addressed reviewer's comments. However, I still believe the theoretical explanation of the CAE is incomplete and must be improved. 
